# Systematic Analysis of a Modified Uni-Traveling-Carrier Photodiode under High-Power Operating Conditions

**Wanshu Xiong** [1,*]**, Zhangwan Peng** [1]**, Ruoyun Yao** [1]**, Qianwen Guo** [1]**, Chaodan Chi** [2] **and Chen Ji** [1]

[1]  The College of Information Science and Electronic Engineering, Zhejiang University, Hangzhou 310058, China
[2]  Research Institute of Intelligent Networks, Zhejiang Laboratory, Hangzhou 311121, China
*  Correspondence: xiongwanshu@zju.edu.cn

**Abstract:** We theoretically analyzed the detailed carrier transport process based on the drift-diffusion model in the InGaAs/InP modified Uni-Traveling-Carrier Photodiode (MUTC-PD) under high optical input power conditions. A high-speed MUTC-PD design was simulated in depth using the commercial simulation software APSYS. The complex interplay between photo-electron and hole transport processes was quantitatively analyzed. The slowdown of hole transit time due to E field reduction in the undoped InGaAs absorber layer dominated the response speed of MUTC-PDs at a high optical power level. The optimized MUTC-PD design has a relatively strong dependence on optical power level. Based on an optimized design, an O–E conversion responsivity around 0.15 A/W and the intrinsic 3 dB bandwidth of 172 GHz were demonstrated when the input optical power density reached 20 mW/$\mu$m$^2$. Our simulation analysis results presented here can be utilized for designing broadband MUTC-PDs in future sub-Terahertz free-space data link applications.

**Keywords:** Uni-Traveling-Carrier Photodiode (UTC-PD); hybrid absorber; drift-diffusion model; structural optimization; high power

## 1. Introduction

Terahertz (THz) wave generation technologies have developed rapidly in the past two decades due to their various applications in THz sensing, security imaging, non-destructive material inspections, and high-speed THz wireless data communications [1–7]. Optical heterodyning, a method realized by photo-mixing the beating signals of two laser modes of different wavelengths, is regarded as a simple and efficient solution for continuous tunable THz wave emission [8]. The Uni-Traveling-Carrier Photodiode (UTC-PD), as the photo-mixer element, is a key component in the optical heterodyning process for achieving an ultrafast O–E conversion in the THz frequency range [9–11]. Recently, a novel modified UTC-PD (MUTC-PD) design with a hybrid absorber section was demonstrated with a faster response speed and high O–E conversion efficiency compared to a conventional UTC-PD design. This modified absorption section consists of a heavy p-type-doped and a lightly n-type-doped InGaAs layer. The lightly n-type-doped InGaAs layer will be fully depleted during device operation under reverse biasing, resulting in an electric-field-assisting high-speed electron transport [12].

Previous researchers have investigated various MUTC-PD designs, mainly under a low optical input power condition, where photogenerated carrier accumulation is negligible [10–14]. However, in real-life free-space THz applications, the optical input power density requirement is much higher to compensate for the strong THz frequency attenuation in the atmosphere [6]. Under these conditions, strong photogenerated electrons and hole accumulation can occur inside the MUTC-PDs and degrade the photo-carrier transport process. Tadao Ishibashi et al. and Xiaomin Ren et al. proposed the optimization of UTC-PDs under high-power conditions by analyzing electron drift velocity performance [15–17]. Joe C. Campbell's Group reported how the cliff layer helps MUTC-PDs attain high saturation currents [18].

Our work systematically investigates how various epitaxial design parameters, particularly the absorption layer, affect the high-speed performance of MUTC-PDs under high optical input power conditions. The accumulated photo-carriers reduce the electric field intensity from the space charge effect inside the device and can cause significant MUTC-PD response speed deterioration [19]. We systematically analyzed the carrier transport process inside a MUTC-PD under high-power conditions using a 3D commercial semiconductor device simulation software. Contrary to the conventional understanding that UTC-PD high-speed response is primarily dominated by the electron transport process alone, our work quantitatively explains the complex interplay of photogenerated electron and hole transport processes inside different regions of the MUTC-PD under high optical power conditions. In particular, the critical roles played by the quasi-E field effect in the p-type absorber section and electron accumulation at the InGaAsP transition layer are quantified. Overall, our simulation results can be utilized in designing high-power MUTC-PDs for ultrafast O–E conversion applications in the THz frequency range.

## 2. MUTC-PD Carrier Transport Qualitative Analysis (Low Optical Power)

A typical MUTC-PD structure operating at a x1550 nm wavelength includes several main elements, including a heavy p-doped InGaAs absorber layer and a lightly n-doped InP electron collection layer. Figure 1 shows a simplified schematic diagram. A thin undoped InGaAsP layer between the absorber and collector layers forms a grading bandgap transition. In a MUTC-PD structure, a part of the absorber will be lightly n-doped, which becomes fully depleted under proper reverse biasing. The photogenerated carriers are accelerated by a strong electric field generated in this depletion region, speeding up the MUTC-PD response when compared to a conventional UTC-PD. The carrier transport process can be described with the carrier drift-diffusion model. In this model, the carrier drift velocity is determined by the carrier mobility and intensity of the applied electric field. An analysis of the MUTC-PD speed response for the low optical power case is given below.

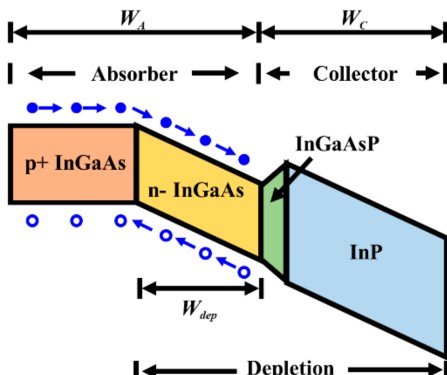

**Figure 1.** The schematic energy band diagram of MUTC-PD shows the absorber, collector, and InGaAsP transition layers. Arrow indicates the direction of photogenerated electron (solid circle) and hole (open circle) movements.

In the heavy p-doped (p+) InGaAs absorber layer, the photogenerated electrons diffuse or drift at a relatively low velocity through this region and then enter the depleted InGaAs absorber region. In contrast, the majority of carrier photogenerated holes relax within a dielectric relaxation time constant [20]. In the depleted absorber region, the photogenerated electrons are accelerated rapidly and reach saturation velocity due to the quasi-ballistic transport process [21]. The movement of photogenerated holes in the depletion layer is in the opposite direction to that of the electrons. The photogenerated hole transport time is determined by holes drifting through the depleted absorber and the rapid relaxation in the p-doped absorber layer. The latter process is considered negligible in time. The overall transit time of electrons and holes in this MUTC-PD structure can be empirically estimated as follows. In Figure 1, let the total InGaAs absorber layer thickness be $W_A$,

the $W_{dep}$ defines the lightly n-type-doped InGaAs absorber layer thickness, and $W_C$ is the electron InP collector layer thickness. In this design, by varying the $W_{dep}$ from 0 to 120 nm, a series of modified absorber structures and corresponding carrier transport time profiles can be analyzed, keeping the total absorber and collector dimensions $W_A$ at 120 nm and $W_C$ at 200 nm, respectively. We assume that both the electrons and holes are at saturation velocity in the depleted absorber region for this calculation.

The hole transit time $\tau_{hole}$ in the depleted absorber region can be roughly estimated as [13]:

$$\tau_{hole} = W_{dep}/v_{hole} \tag{1}$$

$v_{hole}$ is the drift velocity of the hole, which is set to $5 \times 10^6$ cm/s (saturation velocity) in our calculation [22]. The total transit time of electrons includes the transport time $\tau_{diff}$ in the p-type absorber layer and the drift time $\tau_{drift}$ in the lightly n-type absorber layer and collector layer. Each component of transport time can be roughly estimated as follows [23]:

$$\tau_{diff} = (W_A - W_{dep})^2/3De + (W_A - W_{dep})/v_{th} \tag{2}$$

$$\tau_{drift} = (W_C + W_{dep})/v_{electron} \tag{3}$$

$$\tau_{electron} = \tau_{diff} + \tau_{drift} \tag{4}$$

where $De$ is assumed to be 130 cm$^2$/s, a typical electron diffusion coefficient in the p-type InGaAs (p = $2 \times 10^{18}$ cm$^{-3}$). The $v_{th}$ and $v_{electron}$ are the electron thermionic emission velocity in heavy p-type InGaAs and saturation velocity in depleted InP and InGaAs, which are set to $2.5 \times 10^7$ and $4 \times 10^7$ cm/s, respectively [23]. Finally, the total transport time of the electron $\tau_{electron}$ is given as the sum of $\tau_{diff}$ and $\tau_{drift}$.

In Figure 2a, the electron and hole transport time in a MUTC-PD is estimated based on (1)–(4) vs. variation in $W_{dep}$. When the entire absorber is heavy p-doped ($W_{dep}$ = 0 nm), the device becomes a traditional UTC-PD design with its response limited by the transit time of electrons with negligible contribution from hole transport as the absorber is composed entirely of p-type majority material. With increasing depleted absorber section length $W_{dep}$, we find from Figure 2a that the overall electron transport time is reduced with the magnitude of electron diffusion time decrease in the p-type InGaAs absorber exceeding electron drift time increase through the depleted InGaAs absorber and InP collector sections. In contrast, the hole drift time increases in the $W_{dep}$ depleted region and exceeds the overall transit time of electrons when $W_{dep}$ is around 45 nm. At the upper limit of $W_{dep}$ of 120 nm, the entire absorber section is completely depleted, and the device becomes a conventional dual-depletion pin-PD [13]. In this case, the drift time of the hole is over 2.5 ps, leading to a significant slowdown in overall intrinsic response speed. The intrinsic response speed of the MUTC-PD is limited by the slower one among the electron and hole transport processes. The overall carrier transit time $\tau_{tr}$ can be expressed as [24]:

$$\tau_{tr} = \sqrt[4]{\frac{\tau_{electron}^4 + \tau_{hole}^4}{2}} \tag{5}$$

where $\tau_{electron}$ and $\tau_{hole}$ represent the electron transit time and hole transit time, respectively. Finally, the intrinsic 3 dB bandwidth $f_t$ of the MUTC-PD can be estimated with [25]:

$$f_t = 2.4/2\pi\tau_{tr} \tag{6}$$

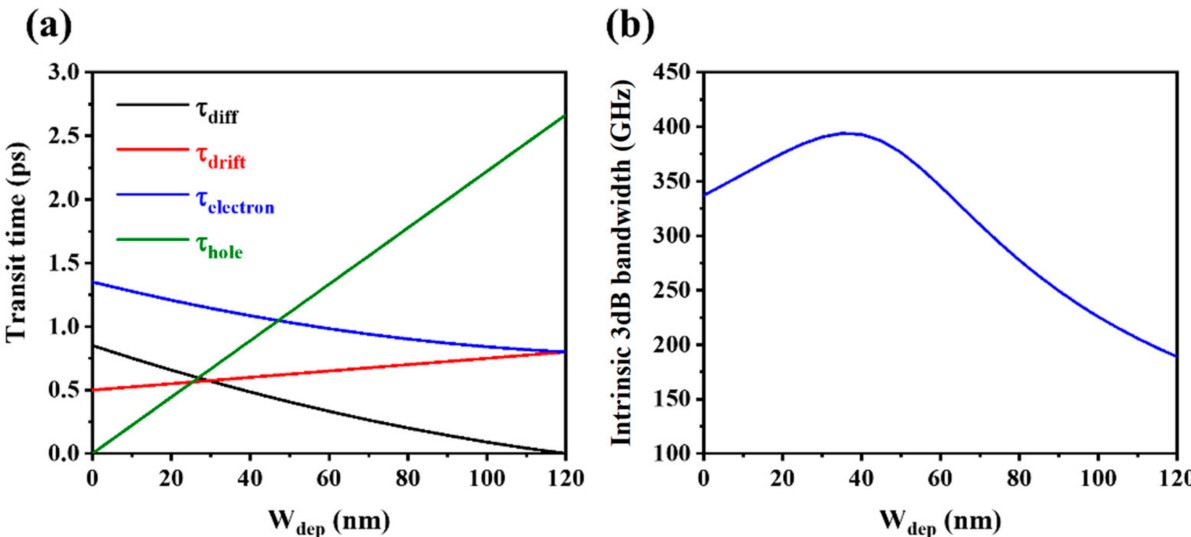

**Figure 2.** (**a**) Calculated electrons and holes transit time in the MUTC-PD as the depleted absorber layer thickness varied from 0 to 120 nm. (**b**) Corresponding intrinsic 3 dB bandwidth calculation results (low optical power condition).

Figure 2b shows the calculated intrinsic 3 dB bandwidth results based on the carrier transit time calculated from (1)–(5) and given in Figure 2a. In Figure 2b, a peak intrinsic 3 dB bandwidth of 394 GHz for the $W_{dep}$ = 45 nm design is estimated from Equation (6). When $W_{dep}$ < 45 nm, the overall carrier transit time $\tau_{tr}$ decreases with increased $W_{dep}$ due to the reduction in the total electron transit time, leading to the rise in the intrinsic 3 dB bandwidth $f_t$. As the $W_{dep}$ exceeds 45 nm, the $\tau_{tr}$ is dominated by the hole transit time $\tau_{hole}$. Increasing $W_{dep}$ will result in longer hole transport time and intrinsic 3 dB bandwidth $f_t$ drop. Overall, an optimal combination of p-type and n-type absorber thicknesses can be found for maximum response speed.

This first-order analysis assumes a low optical input power condition, where the concentration of photogenerated carriers is not large enough to influence the electric field distribution and energy band structure of the device. Nevertheless, our analysis here does demonstrate the effect of $W_{dep}$ on MUTC-PD speed response. It serves as a MUTC-PD design baseline for further systematic quantitative analysis with more detailed epitaxial and device design parameters. In the following sections, we next present our simulation results using a commercial semiconductor device simulator for cases where photogenerated carrier density is non-negligible in the high optical power regime.

### 3. Detailed MUTC-PD Structure

Figure 3a shows the schematic diagram of our bottom-illuminating MUTC-PD design operating at 1550 nm, and Figure 3b presents the detailed epi-structure of the device grown using MOCVD on Semi-Insulating (SI) InP substrate. The topmost layer is a 60 nm thick heavy p-doped InGaAs p-metal contact layer. Below, we have a 15 nm thick p-type $In_{0.6}Ga_{0.4}As_{0.85}P_{0.15}$ layer serving as an electron-blocking layer. Next, we have a 120 nm thick modified absorber layer, including the heavily p-doped InGaAs part and lightly n-doped InGaAs part. The latter will be depleted during device operation. In the following simulation work, we select $W_{dep}$ = 45 nm. Beneath the absorber layer, the electron collector layer consists of a 5 nm thick heavy n-doped InP cliff layer and a 185 nm thick lightly n-doped InP electron collection layer. A 10 nm thick undoped InGaAsP layer is sandwiched between the InGaAs absorber and InP collector layers to form a grading bandgap transition to "smooth" out the abrupt heterojunction interface caused by the bandgap difference. Similar to the p-metal contact layer, a 60 nm thick n-doped InP layer is employed as an n-metal contact layer between the collector layer and Semi-Insulating InP substrate. Finally,

the anode electrode is deposited on the p-metal contact layer and the cathode electrode is deposited on the n-metal contact layer, completing the UTC-PD fabrication process.

**(a)**

**(b)**

| Compound | Thickness(nm) | Doping(cm$^{-3}$) |
|---|---|---|
| In$_{0.53}$Ga$_{0.47}$As | 60 | p: >2×10$^{19}$ |
| In$_{0.6}$Ga$_{0.4}$As$_{0.85}$P$_{0.15}$ | 15 | p: 2×10$^{19}$ |
| In$_{0.53}$Ga$_{0.47}$As | 75 | p: 2×10$^{18}$ |
| In$_{0.53}$Ga$_{0.47}$As | 45 | n: 2×10$^{16}$ |
| InGaAsP | 10 | i |
| InP | 5 | n: 2×10$^{18}$ |
| InP | 185 | n: 2×10$^{16}$ |
| InP | 60 | n: >2×10$^{19}$ |
| S.I InP sub | / | / |

**Figure 3.** Schematic diagram (**a**) and epitaxial design (**b**) of the MUTC–PD device.

The overall MUTC-PD high-frequency response is limited by the combination of carrier transport intrinsic response and device electrical RC parasitic response. Our MUTC-PD has a bottom illumination design with a 20 μm$^2$ active region area and quadruple 50 Ω coplanar lines (an equivalent 12.5 Ω load resistance) in order to minimize the RC parasitic effects [26]. In our design, the calculated RC limited bandwidth is calculated to be 1040 GHz, way beyond the estimated intrinsic response bandwidth from Figure 2. We can, therefore, safely assume the MUTC-PD high-frequency response is completely limited by the intrinsic response related to carrier transport effects and focus solely on the intrinsic response in our simulation.

During the device operation, the 50 Ω load resistance RL is connected to the PD output, and the calculated series resistance RS of our MUTC-PD is 4 Ω, which is small compared to the load resistance. Quadruple 50 Ω coplanar lines (an equivalent 12.5 Ω load resistance) are employed in this work to reduce the effective load resistance. Hence, the corresponding RC time constant is estimated to be (12.5 + 4) Ω × 9.27 fF = 152.96 fs [26], which yields an RC limit bandwidth of f$_{3dB}$ = 1040 GHz. Reducing the active region area is regarded as the effective method to decrease the capacitance of the MUTC-PD and further increase the RC limit bandwidth. Meanwhile, the active region size does not affect the intrinsic response speed of the MUTC-PD. Hence, the intrinsic RC limitation bandwidth is not the limitation of our MUTC-PD device response. We shall focus on the analysis of intrinsic response bandwidth in this paper.

## 4. MUTC-PD Device Simulation Results under Optical Power

We next used the commercial semiconductor device carrier transport simulator Crosslight APSYS to systematically simulate detailed MUTC-PD characteristics, including photoresponsivity, carrier distribution, and electric field profile under high input optical power conditions. A —3V reverse bias is applied to the baseline MUTC-PD design in our simulation, which has a 75 nm p-doped and a 45 nm lightly n-doped absorber layer. The calculated DC photoresponsivity is around 0.15 A/W.

Figure 4a,b shows the simulated intrinsic 3 dB bandwidth vs. optical input power density under applied bias –3 V. When the optical input power density increases from 0 to 3 mW/μm$^2$, the intrinsic 3 dB bandwidth initially also increases. When the optical input power density reaches 3 mW/μm$^2$, a maximum 3 dB bandwidth of 331 GHz is reached. As the optical input power level increases further, the 3 dB bandwidth starts to decrease until it reaches 172 GHz at 20 mW/μm$^2$.

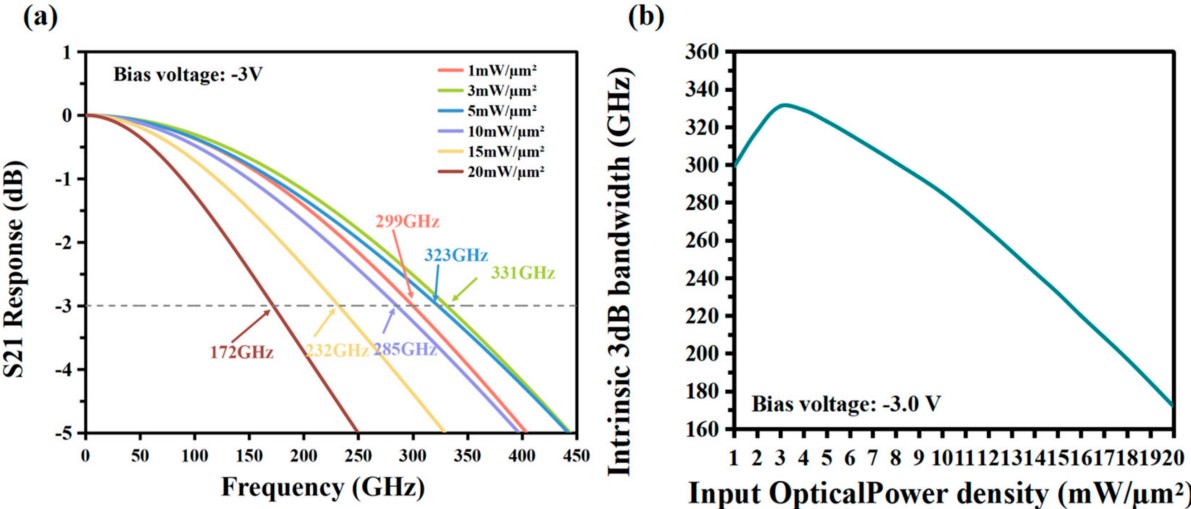

**Figure 4. Simulated results use APSYS**. (**a**) small signal S21 response curves of our baseline MUTC-PD design under different input optical power densities. (**b**) The 3 dB bandwidth plotted vs. optical power density.

The detailed device physics behind the UTC-PD intrinsic bandwidth dependence on optical input power is further investigated by closely examining the photo-carrier and electric field distributions. Under high optical input power conditions, significant photogenerated carriers are accumulated inside the MUTC-PD structure, which modify the internal electrical field and photogenerated electron and hole distribution profiles. The detailed simulation results are shown in Figure 5. Figure 5a shows that the electrical field strength experiences significant reduction with increased optical input power in the InGaAs depleted absorber (0.26–0.305 μm) and InGaAsP transition (0.25–0.26 μm) layers, which strongly impacts the fast carrier drift process. Figure 5b shows the corresponding electron and hole distribution profiles under different optical input power conditions. With increased optical input power density, excess electrons and holes are generated. In Figure 5b, strong photogenerated electron accumulation can be observed in the thin InGaAsP transition layer. Meanwhile, photogenerated hole accumulation can also be observed in the depleted absorber layer close to the p-doped absorber region with a significant positive gradient toward the p-type anode.

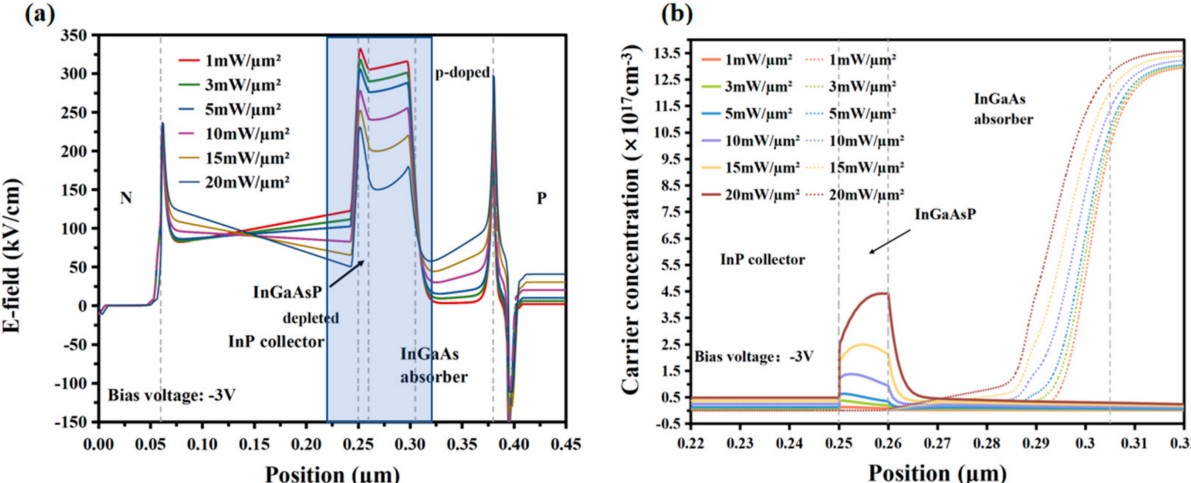

**Figure 5.** (**a**) APSYS–simulated electric field distribution (the direction of the E field is along the positive *x*–axis); (**b**) simulated carrier distribution (solid: electron; dot: hole) for the shaded region in Figure 3a.

Moreover, in the n-type collector (0.06 to 0.25 μm) in Figure 5a, the presence of photogenerated electrons transiting toward the cathode causes a slight negative E field gradient. Still, the average E field level over the collector remains roughly the same and fairly small.

## 5. Discussions

Several effects can be observed in Figure 5 as the optical input power density increases. In the p-type-doped InGaAs absorber layer (0.305–0.38 μm), which remains undepleted, an electric field is nevertheless present, as shown in Figure 5a, with intensity increasing from a negligible level to greater than 50 kV/cm with high optical intensity. This somewhat counter-intuitive effect can be explained by a self-induced quasi-E field, the presence of which is required for meeting the continuity requirement of photogenerated electron and hole currents inside the p-type absorption region [20], and is clearly demonstrated in our simulation here. The intensity of self-induced quasi-E field $E_{ind}$ can also be estimated as [21]:

$$E_{ind} = J_{hole}/qp_0\mu_p \tag{7}$$

Here, $J_{hole}$ is the photogenerated hole current density in the p-type absorber, $q$ is the electron charge, $p_0$ is the p-type doping level of the absorber, and $\mu_p$ is the majority hole mobility in the p-type InGaAs material. Thus, the self-induced quasi-E field effect is enhanced at a higher input optical power level and linearly proportional to the photogenerated hole current. Driven by this self-induced quasi-E field, photo-electrons are swept out of the p-type InGaAs absorber region from the much faster drift process rather than the relatively slow electron diffusion process dominating at a lower optical power level.

Another dominant feature seen in Figure 5b is the strong photo-electron accumulation in the undoped InGaAsP transition layer (0.25–0.26 μm) with increasing optical power. From Gauss's Law, the excess negative charge accumulation results in a negative E field gradient in the InGaAsP transition layer, which can be clearly observed in Figure 5a. Consequently, this negative E field gradient will also reduce the average E field strength throughout the adjacent depleted InGaAs absorber region (0.26–0.305 μm) with increasing optical power density, as also seen in Figure 5a.

In Figure 5a, in the depleted InGaAs absorber, the calculated E field strength is relatively strong, exceeding 150 kV/cm for input power density up to 20 mW/μm². The electron mobility $\mu_e$ in this region (n-type $2 \times 10^{16}$ cm$^{-3}$ doping) exceeds 5000 cm²/Vs [13,22,23]. The product of the electron mobility $\mu_e$ and E field is over $10^8$ cm/s, which is much higher than the electron saturation velocity ($4 \times 10^7$ cm/s). It implies that the electron velocity is already at saturation in this high E field region.

The hole mobility in this depleted absorption layer is about 130 cm²/Vs, far less than the electron mobility. The slower hole drift speed is consistent with the hole accumulation (0.26–0.305 μm) associated with sluggish hole transport, as observed in Figure 5b. With the increase in optical input power, the hole mobility keeps dropping inside the depleted absorber layer. Results in the hole drift do not reach saturated velocity ($5 \times 10^6$ cm/s) at the few hundred kV/cm E field level [21,23–25]. Overall, the reduction in E field intensity in the InGaAs depleted absorber with increasing optical power level does not significantly affect the fast electron drift process but mainly serves to slow down the slower hole drift and results in the deterioration of overall MUTC-PD response speed, as shown in Figure 4.

Finally, as we mentioned in the last section, the average E field level in the n-type collector (0.06 to 0.25 μm) layer remains roughly the same and relatively small. Only very fast photo-electrons need to transit the collector region, and the electron transport time in the collector is not the limiting factor for the overall MUTC-PD response speed.

From the detailed analysis above, we can now quantitatively explain the device physics involved for the intrinsic bandwidth v.s. input optical power in Figure 4. In our bottom-emitting MUTC-PD, the incident optical signal at 1550 nm passes through the transparent InP substrate and the intermediate layers, including the InP collector and InGaAsP transition layer, before being absorbed in the InGaAs absorber, where both

photogenerated electrons and holes generated in the depleted and p-type InGaAs absorber sections can contribute to the overall intrinsic speed response of our MUTC-PD design, dominated by whichever is the slowest transport process across the device.

At a very low optical input level, slow photo-electron diffusion in the p-InGaAs absorber limits device speed. The initial increase in the 3 dB bandwidth with optical power in Figure 4b can be well explained by photo-electrons transiting to the faster drift process in the p-type absorber section caused by a self-induced quasi-E field with an increased photocurrent level. As the optical input power level increases further beyond 3 mW/$\mu$m², the 3 dB bandwidth starts to decrease. The dominant process is now electron accumulation in the InGaAsP transition layer. The resulting sharp negative E field gradient causes strong E field reduction in the depleted InGaAs absorber section, as discussed. The photogenerated holes in this region drifting toward the anode are strongly affected by this E field strength; whereas, photo-electrons operating at saturation velocity are essentially unaffected. Consequently, hole transport slows down due to the reduced E field in the depleted absorber region, becoming a limiting factor for the overall MUTC-PD speed with an increasing optical power level. The transition between these two competing processes can well explain the overall MUTC-PD intrinsic 3 dB bandwidth v.s. optical power results in Figure 4.

Finally, we perform MUTC-PD design optimizations under high optical power input conditions. In Figure 6, we systematically calculate the intrinsic 3 dB bandwidth for our MUTC-PD design, varying the $W_{dep}$ value between 0 and 120 nm with the absorber section (total 120 nm) being partially heavily p-doped and lightly n-doped (depleted) for different optical input power levels. We see from the intrinsic bandwidth v.s. $W_{dep}$ curves that an optimal p-type to n-type absorber segments thickness ratio exists for a given optical power level, representing trade-offs between electron and hole transport processes.

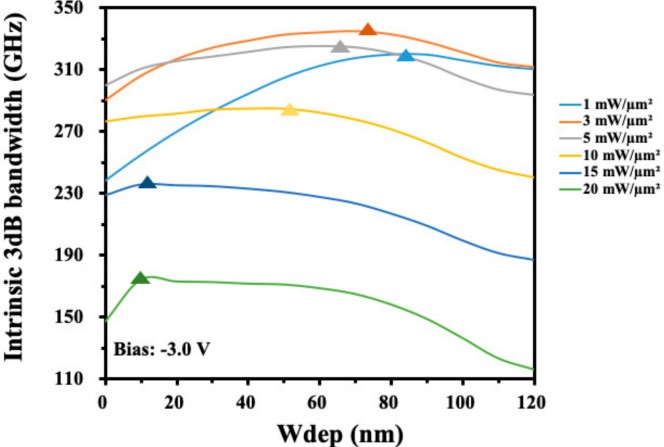

**Figure 6.** Simulated intrinsic 3 dB bandwidth of the MUTC-PD with varying hybrid absorber structures. The thickness of the depleted InGaAs absorber layer *Wdep* varies from 0 to 120 nm, and the optical input power density is set to 1, 3, 5, 10, 15, and 20 mW/$\mu$m², respectively (bias voltage $-$3 V).

Overall, the calculated intrinsic bandwidth in Figure 6 shows a strong dependence on both the $W_{dep}$ design value and the optical input power levels, showing the need for careful device design optimization based on the targeted optical power operating conditions. The optimal $W_{dep}$ is smaller for higher optical power levels, thus minimizing the bandwidth-limiting hole transport time across the depleted absorber section. At 20 mW/$\mu$m², a design with a $W_{dep}$ of 9 nm results in a maximum intrinsic bandwidth of 179 GHz.

## 6. Conclusions

In conclusion, we have systematically investigated the high-speed performance and detailed device physics of an InP-based MUTC-PD design under high optical power

operating conditions. An empirical drift-diffusion model was employed first to qualitatively describe the electron and hole transport processes in a MUTC-PD, which are the main factors determining the MUTC-PD response speed. The effect of photogenerated carrier accumulation in the device under high optical input power conditions was then analyzed quantitatively using commercial simulation software. By carefully analyzing the photo-carrier, E field profiles, and resulting intrinsic bandwidth at different high optical input power levels, we showed that two competing processes determined MUTC-PD intrinsic response at high optical power levels. While the intrinsic bandwidth was limited at a lower power level by photo-electron drift speed in the p-type absorber aided by quasi-E field, at a higher optical power level, strong photogenerated electron accumulation in the transition layer reduced the E field in the depleted absorber section. The device speed was limited by photogenerated hole drift speed through the depleted InGaAs absorber region, which slowed down significantly with increased optical power.

Based on the in depth understanding derived from our work, the MUTC-PD design can be optimized for different operating optical power levels to achieve the highest operating speed, trading off between electron and hole transport time. Overall, our work has broad applications for future MUTC-PD designs in large-scale deployment in free-space THz transmission systems [3–6], where high-power operation is a critical requirement in order to compensate for the strong atmospheric attenuation in this frequency regime.

**Author Contributions:** Conceptualization, W.X.; methodology, W.X.; software, W.X. and Z.P.; validation, W.X., Z.P., R.Y., Q.G., C.C. and C.J.; formal analysis, W.X.; investigation, W.X.; resources, C.J.; data curation, W.X.; writing—original draft preparation, W.X.; writing—review and editing, W.X. and C.J.; visualization, W.X.; supervision, C.J.; project administration, C.J.; funding acquisition, C.J. All authors have read and agreed to the published version of the manuscript.

**Funding:** This work was supported in part by the China National R&D plan 2020YFB1805701 and the Zhejiang Lab grant 2020LC0AD01/001.

**Institutional Review Board Statement:** Not applicable.

**Informed Consent Statement:** Not applicable.

**Data Availability Statement:** Data is unavailable due to privacy.

**Conflicts of Interest:** The authors declare no conflict of interest.

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
