# Peer review of "Systematic Analysis of a Modified Uni-Traveling-Carrier Photodiode under High-Power Operating Conditions"

_photonics, doi:10.3390/photonics10040471_

Round 1

Reviewer 1 Report

In this paper the authors demonstrate a systematic design optimization of a MUTC PD under High power operating condition. This could be from interest for a broad field in the research community. Because of the following points I recommend a major revision:
1. Title: please put the word simulation or CAD in the title since you did not measure anything. You have no separate chapter for optimization just one short section in the discussion. Due to that, a title like “Simulative Design Consideration of a Modified Uni-Traveling-Carrier Photodiode under High Power Operating Condition” would be more appropriate.
2. Page 1 Line 11:
Please omit “for the first time” since you already published the paper “Analysis and Optimization of Modified Uni-Travelling-Carrier Photodiodes under High Optical Power Condition” at 2022 IEEE the 10th International Conference on Information, Communication and Networks.
3. Page 1 Line 39-41:
This is not correct. There are publications which include the high power behavior. For example: Photoresponse characteristics of uni-traveling-carrier photodiodes - T. Ishibashi et al. (2001), Nonlinearities of High-Speed p-i-n Photodiodes and MUTC Photodiodes - G. Zhou, P. Runge (2017)
And so on…
4. Page 2 Line 48-49: See 1.
5. Page 2 Line 67-69
You used the drift diffusion model. Please state what other works have been done with this model. What other models exist? For example the energy balance model. Do you expect any differences between those models regarding the simulation of the high power behavior?
6. Page 4 Line 153 Wrong number. Should be 3.
7. Page 5 Line 180
Please explain how you extracted the value of the series resistance
8. Page 5 Line 181-182
Please write intrinsic RC limitation since you are not including any pads.
9. Page 5 Line 198 wrong number. Should be 4.
10. Page 7 232
Omit for the first time
11. Page 7 234
Should be Jhole
12. Page 7 252-254
You simulated up to 20mW/μm2 and the electron velocity is still saturated. Up to which optical input power is that the case? At what value do you expect the PD to die due to thermal issues?
13. Page 7 257-260
Spelling: and after increase missing, two times the
Please reword since it is difficult to understand. Please try to use shorter sentences.
14. Page 7 272
Spelling: the after MUTC-PD missing
15. Page 7 271-277
Better in section 3
16. Page 7 281
Spelling: the after by missing
17. Page 8 298:
Spelling: optimizations
18. Page 8 299:
Spelling: the before intrinsic missing
19. Page 8 300:
Spelling: the before Wdep missing
20. Page 8 Line 299-304
Reword since bad English and hard to understand. Please try to use shorter sentences.
21. Page 8 Fig 6
In fig. 2 the max 3dB bandwidth is achieved for around 40 nm at low input power. In fig 6 instead for much larger values of wdep. Please give an explanation. And change the y axis in fig 2 also to 3dB bandwidth
22. Page 8 Line 304-308
Omit since repetition
There are more repetitions in the sections 4 and 5. Please avoid.
23. Page 9 Line 315
Wrong number. Should be 6
24. Page 9 Line 316, 322
Omit “for the first time”
25. Page 9 Line 325-330
Reword since bad English and hard to understand. Please try to use shorter sentences

Reviewer 2 Report

This paper theoretically considers the carrier transport process in InGaAs/InP modified Uni-Traveling-Carrier Photodiode (MUTC-PD) under high optical input power conditions. They simulated the electric field distribution as well as carrier distribution under high optical input power conditions for investigating the effect of photogenerated carrier accumulation in the device. Based on their simulation results, the roll-off of 3dB bandwidth at certain input power density is attributed to the slowdown of photo-generated hole drift speed through the depleted InGaAs absorber due to photogenerated electron accumulation in the transition layer. The author has systematically simulated the device physics and optimized the device design under different input power conditions. Their work is worthy of publication in Photonics if they can address the following concerns.

1. Some sentences are ambiguous. The authors need to revise the manuscript carefully.

As seen below,

Line 54 “In particular the important roles played by the p-type absorber quasi-E field effect…”

Line 70” An analysis of MUTC-PD speed response is given below breaking down the contributions from carrier transport of photogenerated electrons and holes…”

Line 186 “To first order the RC-limit bandwidth would not be the limitation of our MUTC-PD device response…”

Line 253 “and should be to first order independent of the observed E field variation at high optical power…”

Line 267 “and to first order the 267 electron transport time in the collector is not the limiting factor for the overall MUTC-PD 268 speed.”

2. It is noticed from Fig. 6 that the intrinsic 3dB bandwidth is not monotonically decreased with increasing input power density when the input power density is no larger than 5 mW/mm2, which has no correspondence to the electric field distribution in Fig. 5. Could you explain this tendency and comment on it?

3. The author picked the total thickness of absorber to be 120 nm and all the following simulations are based on the thickness of 120 nm. Is the choice of 120 nm the optimized condition? Perhaps showing the optimized ratio between Wdep and WA would be more proper for the design guideline. Please comment on the tendency of 3dB bandwidth for the absorber with different thicknesses and different Wdep/WA to provide a more comprehensive guideline.

Reviewer 3 Report

UTC-PDs have been widely used for terahertz generation, and the higher terahertz power requires higher optical input power to the UTC-PD. Previous theoretical analysis and design of UTC-PDs were under the low optical power condition, and this manuscript is the first theoretical work in the high optical power condition to design UTC-PD. The device’s intrinsic speed is determined by both the electron and hole transport processes. The quasi-E field in the p-type absorber reduces the hole transport time at low optical power, and the electron/hole accumulation in the depleted absorber region affects the device’s speed at high optical power.

There are a number of issues or questions that I believe should be addressed before publication.

Major comments:

1) In Line 186 the authors wrote, ”To first order the RC-limit bandwidth would not be the limitation of our MUTC-PD device response”. However, the RC-limit bandwidth is 318 GHz (line 183), which is smaller than the calculated intrinsic 3db bandwidth (394 GHz for Wdep = 45nm). The authors need to clarify more on this.

Minor comments:

1) Line 41: what is the typical optical input power when using the UTC-PD for terahertz generation?

2) Line 33-35: It would be good to put some references of MUTC-PD.

3) Line 182: Where does the “9.27 fF” come from? Please clarify.

4) Line 238-240: It would be interesting to see a discussion about the threshold optical power level where drift process is comparable to the diffusion process.

5) Line 258: It is not clear to me why hole mobility drops due to the hole concentration increases. Can you explain more?

Typo:

1) The equation number between line 233 and line 234 should be Equation (7).

Round 2

Reviewer 1 Report

1. Please carefully revise your paper in terms of scientific English. For example: line 43 has nothing to do with scientific English.

2. Line 58, 236 Please delete also those “for the first time”.

3. Line 189
Where do you get the 12.5Ω from? I see that [26] stated that they used a quadruple 50Ω CPW. Since you want to implement that, too please not just use the value but explain how that works respectively looks like?
